# Sphingosine 1-Phosphate (S1P)/ S1P Receptor Signaling and Mechanotransduction: Implications for Intrinsic Tissue Repair/Regeneration

**DOI:** 10.3390/ijms20225545

**Published:** 2019-11-07

**Authors:** Chiara Sassoli, Federica Pierucci, Sandra Zecchi-Orlandini, Elisabetta Meacci

**Affiliations:** 1Department of Experimental and Clinical Medicine-Section of Anatomy and Histology, University of Florence, Largo Brambilla 3, 50134 Florence, Italy; chiara.sassoli@unifi.it (C.S.); sandra.zecchi@unifi.it (S.Z.-O.); 2Department of Experimental and Clinical Biomedical Sciences “Mario Serio”-Unit of Molecular Biology, University of Florence, Viale GB Morgagni 50, 50134 Florence, Italy; federica.pierucci@unifi.it

**Keywords:** cytoskeleton, extracellular matrix (ECM), mechanotransduction, satellite cells, sphingolipids, sphingosine 1-phosphate receptors (S1PRs), stem cells, skeletal muscle, stiffness, tissue regeneration

## Abstract

Tissue damage, irrespective from the underlying etiology, destroys tissue structure and, eventually, function. In attempt to achieve a morpho-functional recover of the damaged tissue, reparative/regenerative processes start in those tissues endowed with regenerative potential, mainly mediated by activated resident stem cells. These cells reside in a specialized niche that includes different components, cells and surrounding extracellular matrix (ECM), which, reciprocally interacting with stem cells, direct their cell behavior. Evidence suggests that ECM stiffness represents an instructive signal for the activation of stem cells sensing it by various mechanosensors, able to transduce mechanical cues into gene/protein expression responses. The actin cytoskeleton network dynamic acts as key mechanotransducer of ECM signal. The identification of signaling pathways influencing stem cell mechanobiology may offer therapeutic perspectives in the regenerative medicine field. Sphingosine 1-phosphate (S1P)/S1P receptor (S1PR) signaling, acting as modulator of ECM, ECM-cytoskeleton linking proteins and cytoskeleton dynamics appears a promising candidate. This review focuses on the current knowledge on the contribution of S1P/S1PR signaling in the control of mechanotransduction in stem/progenitor cells. The potential contribution of S1P/S1PR signaling in the mechanobiology of skeletal muscle stem cells will be argued based on the intriguing findings on S1P/S1PR action in this mechanically dynamic tissue.

## 1. Introduction

The capability of adult tissues to repair and regenerate after damage strictly depends on the functionality of resident stem cells. In many different tissues endowed with regenerative ability, stem cells mostly reside in a specialized local microenvironment, commonly referred as stem cell niche. Although stem cell niches are distinctive for each tissue, they share many common features, providing complex signals regulating stem cell maintenance and fate [1,2].

The niche includes different neighboring cells, dynamically and reciprocally interacting with the stem cells, and the surrounding extracellular matrix (ECM) [1,2]. In healthy tissues, the niche components essentially act to keep stem cells in a quiescent state preventing their premature differentiation. By contrast, in injured tissues, the niche receives and releases biochemical and physical signals, which contribute to the activation of the stem cells towards their proper functions. In particular, stem cells exit their quiescent state to either proliferate, migrate to the site of injury or differentiate towards the tissue specific cell phenotype, with the aim of morpho-functionally recover the damaged tissues or of self-renewal, thereby ensuring the replenishment of the basal pool of resident stem cells.

A growing body of evidence indicate that ECM is more than a passive support for stem cells, providing instructive signals directing cell behavior [3,4]. Among others, the mechanical properties of ECM, such as stiffness, markedly influence stem cell fate [5,6,7,8,9,10,11,12,13]. The cells are capable of sensing mechanical stimuli from the surrounding microenvironment mainly thanks to the dynamics of several surface components, defined mechanosensors, including, among others, transmembrane receptor, focal adhesion (FA) proteins, transmembrane mechanosensitive channels and mechanosensitive transcriptional factors. Subsequently, they can convert those stimuli into biochemical cascades leading to gene/protein expression [9,14,15,16]. Within such an event, named mechanotransduction, a critical and essential player is the actin cytoskeleton network, which links both physically and functionally the mechanosensors to the nucleus [7,9,17,18,19,20]. Therefore, mechanical properties of ECM and cytoskeleton dynamics are important co-participants in the definition of stem cell behavior/fate. It is worth mentioning that mechanical forces of ECM affect the behavior of stem cells not only in terms of survival, proliferation, migration and differentiation, but also in their ability to modulate, in a reciprocal manner, the deposition, composition, rearrangement or removal of ECM and also to exert force on it. This with the aim of maintaining the desired biomechanical properties of ECM, which enable stem cell proper functionality [3,15,21]. The imbalance of this reciprocal relationship may have dramatic implications, such as either insufficient (chronic) or aberrant fibrotic reparative response.

The identification of the factors and signaling pathways involved in the regulation of mechanobiology of stem cell niche may open new paths for novel therapeutic target in the field of regenerative medicine. In the last two decades, many studies have underlined the relevance of sphingolipids, and in particular, of sphingosine 1-phosphate (S1P) as molecular modulators of the behavior of various cells including stem/progenitor cells as well as of ECM-cytoskeleton linking protein and cytoskeleton dynamic in several cell types.

This review, besides giving an updated description of sphingolipid metabolism and of the signaling mediated by S1P and its specific receptors (S1PRs), focuses on the current knowledge on the contribution of S1P/S1PR signaling in the control of mechanotransduction processes in different cell types including stem/progenitor cells. In particular, the attention will be paid on the role of S1P/S1PR signaling in cytoskeletal dependent phenomena such as FA assembly and lamellipodia formation, implied in cell migration and transduction of signals from plasmamembrane to nucleus, and thus in the control of cell fate. Novel mechanotransducers as targets of S1P/S1PR signaling, namely ezrin, radixin, and moesin (ERM) family proteins and Hippo system, will be also considered. Part of this review will be eventually devoted to the potential contribution of S1P/S1PR signaling in regulating the mechanobiology of satellite cells, the widely regarded resident skeletal muscle stem cells, based on the intriguing findings on S1P action in this mechanically dynamic tissue.

## 2. Sphingolipid Metabolism

Sphingolipids (SLs) are a class of natural bioactive lipids characterized by the sphingoid base backbone, sphingosine (Sph). Since the last three decades, many studies have clearly indicated that SLs are structural components of cell membrane as well as strong bioactive signaling molecules able to control a variety of crucial cellular events, including signal transduction, cell growth, differentiation, and apoptosis [22]. Genetic disorders of biosynthesis of SLs have been described, underlining the relevance of these bioactive metabolites also in humans [23,24].

SL metabolism starts in response to various stimuli [25]. De novo biosynthesis initiates from the condensation reaction of serine and palmitate catalyzed by serine palmitoyl transferase (SPT) in endoplasmic reticulum [26] to produce 3-keto-dihydrosphingosine, which, then, becomes reduced to dihydrosphingosine and subsequently acylated by (dihydro)-ceramide synthase (also named Lass or CerS) [27,28] to form dihydroceramide. Ceramide (Cer), generated by desaturation of dihydroceramide [25] can be then deacylated by ceramidases (CDase) to generate sphingosine [29]. The reacylation of sphingosine that leads to Cer regeneration is named “salvage pathway” [25]. Sph can be also phosphorylated to generate S1P, which in almost all tissues can mediate crucial cellular processes [25,30].

S1P formation is catalyzed by sphingosine kinases 1 and 2 (SphK1 and SphK2) [31], two isoforms differently localized and regulated [32]. Indeed, SphK1 is mainly present in the cytoplasm and after activation mediated by various stimuli, translocates to the cell membrane [32], whereas SphK 2 is located in cell membranes and in organelles, mainly mitochondria and nuclei [33,34,35]. S1P can be reversibly dephosphorylated to Sph by intracellular S1P phosphatases (SPPs) [36] and by extracellular lipid phosphate phosphatases, or irreversibly degraded by S1P lyase (SPL) [37]. S1P degradation by the S1P lyase is the end-point of all SLs. It is worth to underline that due to the presence of a highly integrated network among various bioactive SLs, alterations of one single enzyme or sphingoid molecules such as Cer and S1P, may contribute to changes in the content of the other sphingoids.

### S1P/S1PR Signaling

S1P is a simple membrane-derived lysophospholipid with regulatory roles in almost all processes of mammalian cells. However, the physiological functions of S1P in specific cell compartments, where it is synthetized, are only partially understood. As a signaling biomolecule, S1P is peculiar since it can act through both intracellular and extracellular mechanisms. Intracellularly synthetized S1P can regulate specific targets [38], such as a highly conserved protein that regulates mitochondrial assembly and function, namely PHB2 [39]. Moreover, histone deacetylases 1/2 (HDACs) and telomerase are also some intracellular S1P targets [33,40]. S1P can also be transported outside the cell [41,42] by specific transporters, such as the ATP-binding cassette (ABC) transporters family, Mfsd2b (major facilitator superfamily transporter 2b) expressed in platelets and erythrocytes [43,44] and the transport spinster homolog 2 (SPNS2) [45,46]. SPNS2 represents the master regulator of the secretion of S1P in most mammalian cells exerting a special function in the development and homing of immune cells and in bone homeostasis [47,48].

Extracellularly released S1P can promote pleiotropic biological functions by binding with its specific heterotrimeric G-protein-coupled receptor subtypes, initially named Endothelial Differentiation Gene (Edg) 1-5 and, actually, named S1PR1-5 [49,50,51,52]. Upon ligand activation, S1PRs couple with diverse heterotrimeric G-protein subunits (Gαi, Gαq/11, Gα12/13), leading to different downstream signaling pathways as extensively reviewed in literature [38,53,54]. S1PRs exhibit overlapping or selective expression patterns in various cells and tissues: S1PR1-2 and 3 have a wide tissue distribution, whereas S1PR4 is present mainly in lymphoid tissues, and S1PR5 is expressed predominately in the nervous system. Interestingly, a shift in the content of each different S1PR subtypes has been correlated to various biological processes [55], including skeletal muscle differentiation [56] as well as cell degeneration, such as skeletal muscle cell atrophy [57].

In most mammalian cells, the balance of SPL and SPP activity controls S1P levels. In fact, several in vitro as well as *in vivo* studies reported that SPL inhibition, promoted by genetic and pharmacological downregulation, provokes S1P accumulation [58]. Interestingly, under physiological conditions, circulating S1P levels are significantly higher (10^−7^–10^−6^ M range) in peripheral blood than in solid tissues due to the release of S1P by several blood cell types [44,59,60,61,62] and to the lack of SPL expression in platelets [63] and of SPL and SPPs expression in erythrocytes [64]. In peripheral blood, S1P binds to albumin and apolipoprotein M and circulates as a part of high-density lipoprotein particles. On the other hand, the degradation of S1P by SPL at tissue level contributes to the low level of the bioactive lipid outside of the bloodstream [65].

Notably, the difference in the concentration levels of S1P in blood and in the tissues drives the migration of immune cells [47] as well as of stem/progenitor cells [66]. S1P acting as chemoattractant, is responsible for the attraction of immune cells and their exit from lymphoid organs to circulation and for the passage of the bone marrow progenitor cells from peripheral tissues to the lymphatic system [67]. These S1P functions appear physiologically relevant in the control of the immune system during inflammation as well as in the *in vivo* physiology of vascular systems. Among the various S1PR subtypes, the S1PR1 expression is the critical factor that regulates sensitivity to circulating S1P. In fact, abrogation of S1PR1 expression prevents lymphocyte egress and reduces inflammation [68]. On the contrary, S1PR2 antagonizes migration elicited by chemokines, contributing to relegate immune cells within the tissues [69].

## 3. S1P/S1PR Signaling and Mechanotransduction

Mechanical forces, generated from the ECM environment, drive biochemical signals and molecular interactions resulting in actin cytoskeleton and cell membrane remodeling. In particular, the progressive and successive cycles of cell adhesion, contraction and retraction mediated by movement-associated membrane protrusions, including among others lamellipodia, are the consequences of mechanotransduction events that control cell movement and cell shape. In fact, mechanical stimulation of distinct adhesion proteins contributes to their conformational changes and membrane modifications that, in turn, promote the recruitment of other scaffolding proteins. These events lead to the maturation of nascent FA complexes, premise of the cell migration.

Similarly, mechanical cues promote structure changes of some proteins, such as talin, crucially involved in the signaling transduction upstream to gene expression regulation, thus controlling the cell fate determination [70,71].

### 3.1. Impact of S1P/S1PR Signaling in Cytoskeleton Remodeling/Dynamics for Cell Migration

Emerging evidence indicate that S1P signaling and the S1PR expression profile, are crucially implicated in the movement-associated membrane transformation (i.e., cell adhesion structure variations, lamellipodia formation etc.) as well as in cytoskeleton remodeling leading to cell migration in response to stimuli from the ECM. The conversion of mechanical forces to biochemical signals requires several structures including the FA complexes, which are organized around of specific receptor proteins, the integrin family proteins, binding to ECM and to actin-coupled complex functioning as anchor proteins. Some evidence seems to indicate that integrins can also act as mechanosensors [72].

#### 3.1.1. Focal Adhesions

The first step of cell migration is the dynamic change of FAs. In response to mechanical tension, the recruitment of proteins, such as vinculin and α-actinin, is responsible for size increase of FAs [73]. Successively, FAs tightly associate with the end of stress fibers, which are contractile structures crucially implicated in pushing the cell body [17] accompanied by disassembly of FAs and cell retraction at the trailing edge. The role of S1P/S1PR1 signaling in FA formation has been described in fibroblasts several years ago. In particular, it has been shown that S1P/S1PR axis is required for platelet-derived growth factor (PDGF)-induced FA formation and activation of FA kinase (FAK)/Src and p38 MAPK [74]. S1P-augmented fibroblast chemotaxis via Rho-associated protein kinase (ROCK) activation appeared to be also mediated by S1PR2 activation [75].

S1P through its receptors acts mainly by promoting the co-localization of cortical actin, stress fibers, FA complexes and FAKs [76]. From a molecular point of view, it has been documented that S1P is able to induce tyrosine phosphorylation of FAK, leading to FA disruption and cell-to-matrix and cell-to-cell adherens junctional complex remodeling [77,78] as well as to tyrosine phosphorylation of paxillin, a multifunctional and multidomain FA adaptor protein [79]. However, studies specifically addressing the role of the other S1PR subtypes in FA formation are still scanty.

Although the role of the interaction between integrins and ECM and the their role in mechanotransduction is beyond the purpose of this review, it is important to underline that very recently it has been reported a functional cross-talk between integrin co-receptor CD98hc (SLC3A2) and SL metabolism, which contributes to cell mechanical homeostasis [80]. CD98hc, a transmembrane protein that associates with integrins β1/β3 without any effect on their activation, can regulate integrin-mediated mechanosensing by controlling the level of the delta-4-desaturase (DES2), the enzyme that converts D-erythro-sphinganine to D-erythro-sphingosine. In fact, the depletion of CD98hc decreases the expression of DES2 and reduces SL availability, thus preventing correct membrane remodeling and RhoA-mediated signaling activation, and in turn leading to stiffness sensing impairment.

Another interesting role is played by SLs in endothelial cells (ECs). In particular, in this type of cells, S1P and other factors, such as the hepatocyte growth factor (HGF), can act as barrier-enhancers [81,82]. The relevance of S1P/S1PR system is confirmed by the findings that impairment in S1P signaling and barrier integrity disruption has been observed in pathological conditions, such as several pulmonary disorders [83]. The protective action of S1P on blood vessels is the result of S1PR activation and the subsequent intricate crosstalk with other signaling molecules, such as MAPKS and ROCKs. The EC barrier enhancement has been mainly attributed to the activation of S1PR1/Rac pathway. Contrarily, the binding of S1P to another S1PR subtype, S1PR3, promotes barrier disruption and loss of tight junction formation. Additional studies demonstrated that the signaling promoted by S1PR3 leads to vasodilation through endothelial nitric oxide synthase activation in arteries [84]. Differently to S1PR1, S1PR2 gene disruption promotes the decrease of vascular permeability by affecting endothelial tight junctions, indicating a positive role of the signaling triggered by this receptor subtype [85,86,87]. The dissimilar functions of the S1PRs are due to the specific coupling with different G-proteins: S1PR1 couples solely with G_i/o_, whereas S1PR2 and S1PR3 also couple with G_12/13_ inducing the stimulation of RhoGTPase, and, in turn, cortical actin destabilization, stress fiber formation, and endothelial barrier disruption. It has also been reported that in the microvascular endothelial model, the bioactive lipid may regulate the EC barrier permeability through S1PR1 as well as S1PR2 signaling [88]. On the other hand, a role for S1PR5 has been proposed in ECs localized in the central nervous system, where its silencing results in the reduction of leukocyte migration. This finding suggests that S1P/S1PR5 signaling may control barrier disruption and permeability during inflammation in the nervous system [89]. Interestingly, a recent clinical study reported that in patients with SphK1-positive colorectal cancer, S1P formation is correlated to enhanced cell migration and metastasis with a worse prognosis with respect to patients that are affected by SphK1-negative cancer [90]. As demonstrated in this study [90], SphK1/S1P may control through S1PR signaling the expression of phospho-FAK, p-protein kinase B (AKT) and cytoskeleton reorganization.

#### 3.1.2. Lamellipodia

The first evidence of a role of S1P in lamellipodia formation was reported by Stam et al. [91]. In this report, it was shown that T-lymphoma invasion is due to LPA/S1PR-mediated RhoA and phospholipase C signaling pathways, which lead to pseudopod formation and enhanced infiltration. In the last two decades, the regulation of membrane protrusion formation by S1P/S1PR signaling has been shown in many circumstances and cell types [92,93,94]. Because of the difference in the expression of S1PR subtypes coupled to distinct GTP-binding proteins, S1P may induce opposing effects, either stimulatory or inhibitory, on cell migration and homing of various types of progenitor cells [95], such as bone marrow derived-mesenchymal stromal/stem cells (MSCs). In particular, it has been shown that the receptor subtypes, S1PR1 and S1PR3, are responsible for S1P-induced migration of human MSCs, whereas S1PR2 mediates its inhibition, activating Rho/ROCK system [96]. The bioactive lipid potently stimulates motility of ECs [97] and migration of keratinocytes [98], suggesting a positive role of the bioactive lipid on cutaneous wound closure. S1P is also a potent cell migration inducer for some tumor cells, such as glioma cells [99,100]. Contrarily, S1P exerts inhibitory effect on the melanoma, fibrosarcoma [101] and breast cancer cell migration [102].

Recently, it was reported that the formation of complexes between proteins involved in SL metabolism and membrane modifications is crucial for cell migration [103]. In fact, on the lamellipodia side, SphK1 and the receptor S1PR1 form a complex with filamin A, an actin-binding protein that is known to cross-link cortical actin filaments at membrane ruffles. In line with this, other novel findings indicate that the cytoskeletal components, such as cortactin and non-muscle myosin light chain kinase, assemble with S1PRs. In particular, this event occurs in caveolin-enriched microdomains (CEM), where integrin β4 forms a complex with S1P-activated S1PR1 [104], suggesting the importance of protein-protein interactions and localized S1P production. Furthermore, it is worth to note that a specific membrane localization and protein assembly of S1PRs not only contribute to lamellipodia formation/cell migration, but also permit specific signaling pathways. In fact, it has been recently demonstrated that extracellular α-Synuclein can provoke mislocalization of the receptor subtype S1PR1 out from CEM. This event induces the uncoupling of S1PR1 from G_i_ protein that, in turn, leads to the increase in α-Synuclein content. Therefore, the impairment in S1PR1 localization/signaling may contribute to the protein accumulation and aggregation, characteristic features of idiopathic Parkinson’s disease [105].

Based on the relevance of released S1P and its action through S1PR signaling on membrane protrusion formation and remodeling, cytoskeleton rearrangement and cell migration, several therapeutic strategies have been developed [106]. In particular, clinical and preclinical studies have reported that S1P analogues may be useful as protective factors especially in inflammation and autoimmune disorders. Fingolimod (FTY720) is the first S1PR modulator clinically approved as first-in-class drug targeting S1PR. Although the clinical relevance of Fingolimod is clear, the mechanism of action of this drug is not fully clarified. It is known that it is phosphorylated *in vivo* after ingestion. In lymph nodes, the drug initially activates S1PR1 and then induces its internalization. Thus, the cells appear less sensitive to the S1P gradient and remain within the lymph node [107,108]. S1PR1 agonist/antagonists are mainly exploited in pluripotent multilineage hemopoietic cells. The antagonist of S1PR1, W146, significantly stimulates the *in vivo* mobilization of Kit^+^/Sca-1^+^/Lin^−^ hematopoietic stem progenitors from bone marrow into the blood, suggesting the involvement of this receptor subtype in keeping hematopoietic cells inside the origin microenvironment [109]. In addition, since hematopoietic stem and progenitor cells present higher levels of S1PR3 in comparison to differentiated cells, and this receptor subtype is crucial for niche localization, it has been demonstrated that the pharmacological antagonism (e.g., VPC01091) or the knockout of S1PR3 mobilizes these cells into blood circulation [110]. The role of S1PR2 in neural progenitor cells has also been determined by Kimura et al. [111]. The authors demonstrated that the blockade of S1PR2 by JTE-013 improves the migration of these progenitor cells in response to a S1P gradient. It is worth to note that the S1P analogues not always mimic the effects of endogenous S1P. For instance, S1P and FTY720 act differently in promoting EC barrier enhancement: only S1P promotes calcium increase and S1PR1 phosphorylation, whereas phospho-FTY720 does not induce Rac1 activation or cortactin phosphorylation [112]. Interestingly, other more selective S1PR modulators have been synthetized and subjected to clinical trials [113]. For example, FTY720-phosphonate has been shown to activate Rac1, induce FA formation and redistribution of actin and cortactin in cell periphery of lung ECs, mimicking S1P effects [114]. In line with these data, the treatment with a novel S1PR5 antagonist, BIO-027223S1PR5, demonstrated that S1PR5 is fundamental for human natural killer cell migration and for the exit of these cells from the bone marrow to the blood circulation [115].

### 3.2. Impact of S1P/S1PR Signaling in Cytoskeleton Remodeling/Dynamics for Cell Fate

Important physiological adjustments in cell adhesion, cytoskeleton remodeling and functional protein interactions are crucial during cell differentiation in response to mechanical properties of ECM [6,7,11,116,117].

Cytoskeleton dynamics and rearrangement of its components, such as actin filaments, arise when stem cells start to differentiate into cells of a specific lineage. For instance, few FAs and dispersed actin filaments facilitate the adipogenic differentiation of MSCs, whereas higher number of FAs and well-organized cytoskeleton promote their commitment into osteoblasts [118]. It has also been reported that disrupting actin depolymerizing factors (i.e., Cofilin 1) reduces the ability of MSCs to undergo adipogenesis, while the treatment with the actin polymerization inhibitor, such as Cytochalasin D, potentiates cell differentiation, confirming the existence of a link between cytoskeleton organization and MSC lineage commitment [119].

The meaning of this biomechanical regulation for the tissue repair/regeneration is also proven by the pathological consequences of ECM/cytoskeleton dysregulation. In fact, when the mechanical properties of tissue are impaired, the effects are considerable: mutations in the genes encoding intermediate filament proteins are associated with pathologies even in humans, including amyotrophic lateral sclerosis [120,121]. In addition, mutations in desmin, filamin C, etc., increase the stiffness and limit the tolerance to mechanical stress in skeletal muscle fibers leading, in some cases, to myofibrillar myopathies [122]. Changes in mechanical properties may also lead to cell transformation. It was, in fact, reported that altered tissue stiffness could disturb morphogenesis and guide epithelial cells towards a malignant phenotype [123].

Several lines of evidence underline the role of S1P/S1PR signaling in the modulation of MSC differentiation via cytoskeleton remodeling. For instance, the inhibition of the S1PR2-mediated signaling, leading to ERK phosphorylation and actin rearrangement, determines MSC clonogenicity and migration, while decreasing cell differentiation into adipocytes and mature osteoblasts [96]. It is known that S1P/S1PR signaling can also increase the expression of cardiac lineage and smooth muscle markers in c-kit^+^ cardiac progenitor cells. In particular, it seems that S1PR2 and S1PR3 expressed in cardiac progenitor cells can trigger Gα_12/13_/RhoA signaling affecting the commitment of these progenitor cells. Therefore, the downstream signaling to these S1PR subtypes may be crucial in favoring the tissue response to injury [124]. In addition, very recently, a study performed in osteocytes have established the role of S1P metabolism in the control of bone mass and architecture when the cells are subjected to interstitial fluid flow [125]. In fact, a downregulation of S1P transporter SPNS2, and of the enzymes responsible for degradation and dephosphorylation of S1P, occurs in response to this mechanical loading, thus suggesting a functional link between S1P metabolism and cell mechanotransduction [125]. It is worth to note that the importance of the biomechanical properties of ECM in the process of cell differentiation has been underestimated and misunderstood by culturing the cells on stiff substrates, with traditional methods. For instance, Berdyyeva et al. [126] reported that primary epithelial cells cultured in plastic dishes increase in the stiffness with passaging. Similarly, when epithelial breast cancer cells and endometrial adenocarcinoma cells are cultured on glass and plastic surface, respectively, were found to stiffen and express more α-actin, confirming the influence of the culture surface and conditions on cell fate decision [127].

Although it is beyond the scope of this review, here, we only underline the importance to consider carefully the crucial point of cell culture conditions especially in order to better understand the spatio-temporal relations between cells, matrix and soluble factors, such as S1P, in the three-dimensional niche. Furthermore, because cytoskeleton has a role in mechanotransduction and its structural components are very dynamic, it can be speculated that the cytoskeleton may be a “memory storage of cell shape”. Interestingly, it is known that actin structures remain after mitosis and some stress-fibers persist in daughter cells [128].

Further research is needed to better understand how the signaling starting from plasmamembrane can contribute to conserve the cytoskeleton organization and influence cell fate. Interestingly, as stated more extensively in the Section 4.2, a correlation between cell shape changes in response to mechanical forces and gene expression regulation has been proposed to occur via the Hippo system, which is also a specific mechanotransducer target of S1P /S1PR signaling.

## 4. Mechanotransducers as Targets of S1P/S1PR Signaling

As stated above, the simultaneous expression/activation of several S1PR subtypes in one cell and the coupling of each subtype to various G-proteins lead to specific and complex responses [129,130]. Several mediators of mechanical inputs are also effectors in the multiple signaling pathways triggered by S1P/S1PR system. Thus, focusing on the specific object of this review, here we consider only some novel mechanosensors/transducers as targets of S1P/S1PR signaling: the ezrin, radixin and moesin (ERM) protein family [131] and the Hippo system [14,132].

### 4.1. ERM Protein Family

Ezrin, radixin and moesin are the components of ERM protein family [133]. They are scaffolding proteins able to link directly the plasmamembrane to the cortical actin through Rho GTPases [131,134]. By mainly interacting with cadherins and integrins, ERM proteins contribute to the communication cell-ECM as well as cell-neighboring cells, thus acting as mechanotransducers. Moreover, ERM proteins are able to act as anchor proteins for localizing proteins, close to their targets (i.e., protein kinase A) and facilitating membrane transport of electrolytes through the regulation of ion channels and transporters [135]. Thereby, ERM proteins play an important role in the regulation of several biological processes such as cell morphology [134], cell adhesion [136] plasmamembrane protrusion formation and, thus, cell migration [137]. It has been also documented that ERM protein family can influence stem cell differentiation by regulating cell stiffness and reorganizing FAs. For instance, it has been reported that ERM knockdown determines reduced actin disassembly and cell stiffness, thus leading to a time-dependent impairment of adipogenesis process [138]. Notably, ERM protein family dysfunctions are also associated with several cancer types, such as osteosarcoma [139] and breast cancers [140].

The activation of ERM proteins is controlled by conformational changes. When activated, the protein structure shifts from the inactive state, characterized by cytosolic localization, to the active one, marked by plasmamembrane localization due to the phospatidylinositol 4, 5-bisphosphate binding at amino-terminus of the protein and the phosphorylation of the carboxyl-terminus. In the literature, few protein kinases are described to be able to phosphorylate ERM proteins *in vivo*, such as PKC, G protein-coupled receptor kinase 2 and the myotonic dystrophy kinase-related Cdc42-binding kinase [141].

Importantly, several reports indicate that SLs and, in particular S1P, can regulate ERM family protein activation in many cell types, including tumor cells, such as breast cancer and glioblastoma cells [142] as well as chondrogenic cells [143]. The first evidence of the role of SLs in ERM protein functions was obtained by Zeidan et al. [144]. The authors demonstrated that the disruption of the acid sphingomyelinase/Cer pathway contributes to prevent cisplatin-induced cytoskeletal changes and ezrin dephosphorylation. In another study, S1P, acting through S1PR2, was found to be able to promote strongly the C-terminal phosphorylation of the ERM proteins in HeLa cells transfected with recombinant bacterial sphingomyelinase, influencing actin cytoskeleton remodeling and filopodia formation [145]. Similarly, in human embryonic fibroblasts, the S1P/S1PR2 signaling is important to regulate cellular architecture through the modulation of ERM protein phosphorylation [146]. In HeLa cells, it has been documented that S1P, specifically formed by SphK2 and not by SphK1 activity, is essential for cell invasion elicited by epidermal growth factor [147]. In fact, the increased S1P production, achieved by overexpression of SphK2, likely in a specific cellular compartment, is sufficient in promoting ERM protein activation. SphK/S1P system via ERM protein activation has also been shown to regulate phosphate-induced vascular smooth muscle cell matrix mineralization, suggesting new potential therapeutic approach for inhibiting or delaying vascular calcification [148]. ERM proteins are also modulated by S1P in human pulmonary ECs, where they contribute to the function of S1P as barrier-enhancer [149].

Interestingly, in this study, it is shown that, although the ERM proteins are structurally similar, each one can differently affect the modulation of S1P response, in terms of cytoskeleton remodeling and permeability. In T cells lacking moesin, S1P fails to promote the S1PR1 internalization and clathrin-coated vesicles formation, supporting the importance of this specific protein as scaffold. Furthermore, when these moesin-deficient T cells were treated with FTY720 a delay in lymphopenia was observed, indicating a distinct function of this protein in regulating S1PR1-mediated signaling [150]. Moreover, S1P promotes the phosphorylation/activation of ezrin [146,147] and in a PKC-dependent manner, the bioactive lipid stimulates pulmonary ECs via the activation of ezrin and moesin, but not of radixin [151].

### 4.2. Hippo System

Emerging evidence indicate that the mechanical properties of ECM can be transduced from the plasmamembrane to the nucleus by a direct mechanocoupling through cytoskeleton [152,153,154]. Indeed, it has been reported that actin cytoskeleton disruption, promoted by pharmacological or genetic approaches, can induce changes in nuclear morphology, lamin A/C expression, chromatin dynamics, histone post-translation modifications and then in gene transcription [155]. In these mechanobiological responses, the nuclear envelope proteins can play a central role [156]. It has been described that ECM stiffness acts as upstream regulator of a signaling pathway named the Hippo pathway [132], which was firstly described in Drosophila in a genetic mosaic screen finalized to the identification of genes involved in the growth of larval tissues [157].

The main components of the Hippo system in mammalian cells are the transcription factor YAP (Yorkie homologous Yes protein-tyrosine kinase-associated protein), the co-activator TAZ (PDZ-binding motif) and their upstream regulators represented by the tumor suppressive kinases Mst1/2 and Lats1/2 [132].

The role of YAP/TAZ as mechanotransducers of the information either from cell shape or from the ECM to the nucleus are reported in several studies [14,158,159,160]. In particular, ECM stiffness controls the kinase cascade and, in turn, the shift between the active/inactive state of YAP. When the cell is round-shaped, due to less stiffer or smaller ECM adhesive area, the tyrosine-phosphorylation of YAP facilitates its cytoplasmic localization and its degradation by proteasomes. Therefore, when YAP/TAZ is out of the nucleus, the gene transcriptional activity is off, cells arrest in G0 and enter the commitment program. Contrarily, when cells are plated on large and stiff substrates, the Hippo pathway is inhibited due to the high cytoskeletal tension mediated for example by ROCK. Therefore, when YAP/TAZ, are in the dephosphorylated state, accumulate into the nucleus promoting cell proliferation and inhibiting cell differentiation [14].

Similarly, the contact inhibition promotes YAP/TAZ phosphorylation and cytosolic localization. On the other hand, at low density, cells divide and active YAP/TAZ localizes into the nucleus, functioning as pro-proliferative transcriptional factors. Target genes of YAP/TAZ activity including connective tissue growth factor (CTGF) have been described [14,161].

Recent studies demonstrated that the stimulation of some G protein-coupled receptors (GPCRs), such as S1PRs, controls YAP activity [132]. It has been shown that S1P/S1PR2 signaling determines a strong dephosphorylation and activation of YAP/TAZ leading to CTGF expression and proliferation increase in hepatocellular carcinoma cells [162]. S1PR2-mediated signaling activates YAP proteins in ovarian cancer cells favoring their proliferation [163,164]. Interestingly, HEK 293T cells transfected with the S1PR2 (R150H) mutant show receptor retention into intracellular vesicles, the block of G-protein coupling and low Hippo activity [165].

However, it is worth to note that, in several circumstances, S1P/S1PR signaling seems to be independent from the Hippo pathway kinases. In fact, it is known that the bioactive lipid also induces YAP nuclear localization through the G_12/13_- or G_q/11_-coupled S1PR2, Rho GTPase activation and F-actin polymerization [163]. Moreover, by bioinformatic analysis, it has been demonstrated that YAP signature gene expression panel is significantly higher in tissue samples obtained from S1P lyase-deficient mice, indicating that the accumulation of S1P may regulate the YAP function *in vivo* [163]. In the same study, it has been also shown that serum starvation of HEK293A cells disrupts actin stress fibers and forces YAP into the cytoplasm. Interestingly, these events can be counteracted by S1P treatment. Notably, another SL, sphingosylphosphorylcholine, affects YAP activity [166]. Since YAP/TAZ are important regulators of the cell cycle, they play a crucial role in stem cell determination and tissue regeneration. For instance, high stiffness, elevating YAP/TAZ, determines bone differentiation of MSCs, whereas a lesser tough environment inactivates YAP/TAZ and allows cell commitment into other cell lineages, such as adipocytes [14]. Interestingly, YAP/TAZ activity is repressed when the integral type I endopeptidase MT1-MMP (Membrane type I-matrix metalloproteinase) is knockout, preventing MSC differentiation in osteocytes, thus, leading to osteopenia [167]. Despite the scanty evidence, based on the relationship between actin cytoskeleton, YAP/TAZ activity and S1PR-mediated signaling, important consequences on stem cell fate can be expected. In fact, the YAP target genes are transcripted in human embryonic stem cells after treatment with exogenous S1P [168]; the treatment with the bioactive lipid is able to promote follicle growth by increasing nuclear YAP, thus, suggesting a potential role for S1P analog in polycystic ovarian syndrome treatment [169]. Moreover, S1P can regulate cardiac precursor cell migration by controlling YAP signaling, playing a role in endoderm formation in organism model Zebrafish [170].

Based on this emerging evidence that YAP is a downstream effector of some biological functions of S1P, further investigation of the crosstalk between YAP/TAZ and S1P/S1PR signaling may open new insights on the functional relationship of these two pathways.

## 5. Mechanobiology of Skeletal Muscle Stem Cell Niche: The Potential Role of S1P/S1PR Signaling

Several studies in the last years underlined the role of S1P/S1PR signaling pathway in skeletal muscle biology, including resident skeletal muscle myogenic precursor/satellite cell proliferation, differentiation and migration. By contrast, to date, no data are available on the involvement of this signaling pathway in the regulation of mechanobiology of satellite cell niche namely: (1) satellite cell mechanotransduction events in response to ECM stiffness and (2) modulation of ECM stiffness. However, it appears intriguing to postulate this involvement when considering that: (1) skeletal muscle tissue is highly mechanically dynamic and, therefore, satellite cells are expected to experience strong mechanical forces during each routine contraction-relaxation cycle and, also, to gauge the intrinsic mechanical properties of surrounding ECM; (2) experimental evidence addressed the role of S1P/S1PR signaling as molecular modulators of ECM cytoskeleton-linking protein and cytoskeleton dynamics in satellite cells as well as of the functionality of different cells involved in ECM building.

### 5.1. Satellite Cell, Interstitial Cell and ECM Dynamic Interplay

It is well accepted that skeletal muscle tissue possesses a remarkably intrinsic ability to regenerate after focal damages, mainly thanks to the activity of the satellite cells, widely regarded as the resident muscle stem cells [171,172].

In healthy skeletal muscle, satellite cells reside mitotically quiescent and transcriptionally inactive in an unique and specialized anatomic microenvironment at the periphery of differentiated myofibers (hence the name) in intimate association with the myofiber sarcolemma underneath the basal lamina, known as satellite cell immediate niche [173]. By contrast, in injured muscle, satellite cells exit the quiescent state becoming activated to execute the myogenic program, reminiscence of what occurs during embryogenesis, culminating in the formation of new polynucleated myofibers capable to morpho-functionally recover the damaged tissue [174]. There is evidence suggesting that, in parallel, a small percentage of satellite cells undergo to self-renewal thereby ensuring the replenishment of the basal pool of resident satellite cells, ready for responding to future demands [175].

As for other stem cells in different tissues, the functionality of satellite cells is strictly dependent on the features of the surrounding microenvironment, which, beside the immediate niche, includes the microenvironment beyond. The latter may be identified by the interstitial space between myofiber, where different interstitial cells, motor neuron endings, vascular network with associated secretable factors are located, embedded in the ECM. Moreover, the systemic milieu has to be included, comprising molecular and cellular signals deriving from the entire muscle belly along with neighboring skeletal muscles and bones, immune cells and circulating factors including among others gastrointestinal tract hormones [173,176].

Growing evidence indicated that critical factors for the regulation of satellite cell activation and myogenic differentiation are represented by: (1) juxtacrine and paracrine interactions that satellite cells establish with some resident or migrated interstitial cells [171]. In particular, a role in supporting satellite cell-mediated regeneration has been documented for interstitial cells, such as pro-inflammatory phagocytic macrophages (M1) and anti-inflammatory pro-regenerative macrophages (M2), fibroblasts, myofibroblasts, different MSCs including fibro-adipogenic progenitors (FAPs), telocytes and capillary endothelial and periendothelial cells including perycites [171,177]. (2) Intrinsic biomechanical features of ECM embedding the cells such as stiffness [178,179,180,181,182].

ECM of satellite cell microenvironment includes essentially the basement membrane and the overlying endomysium sheath, intimately linked to it, surrounding each myofiber to form the interstitial stroma. The basement membrane is composed of two layers: one internal, linked to the sarcolemma, called basal lamina, wrapping the underlying satellite cells, and a second, external, namely fibrillar reticular lamina. The main ECM components are represented by non-fibrillar type IV collagen, fibrillar type I and III collagen closely assembled to form collagen fibers, assorted glycoproteins, mostly serving as cell adhesion molecules, embedded within an amorphous proteoglycan-rich ground substance. For a more detailed description of components of ECM surrounding satellite cells, we refer to the following reviews [178,183,184].

The content in fibrillar collagen is the main determinant of ECM stiffness. Such property measures the resistance to deformation; in other words, it may be defined as the resistance of collagen fibers to breakdown under mechanical forces during stretching [12,15]. Several *in vitro* studies showed that myogenic differentiation of satellite cells requires an optimal ECM stiffness and that either lesser or more stiff coatings negatively affect the ability of myogenic cells to proliferate and differentiate [179,185,186,187]. An interesting study showed that collagen VI null murine muscles display a significant decrease in stiffness, which correlates with an impairment of the *in vitro* and *in vivo* activity of satellite cells [188]. In addition, it has been demonstrated that the impairment of satellite cell functionality and reduced regenerative capacity in aged or pathological muscles is associated with an increase of intrinsic ECM stiffness [182,189,190,191,192,193].

It is clear that ECM stiffness is related to the activity of the embedded cells, responsible for the constant collagen deposition, degradation (by secreting different MMPs) organization/reorientation or cross-linking. These cells are mainly represented by resident interstitial fibroblasts and myofibroblasts [15,194]. However, many other cell types present in the satellite cell microenvironment may contribute to ECM turnover/stiffness including the same satellite cells, interstitial MSCs and infiltrating inflammatory cells [195,196].

Satellite cells may employ different mechanosensors to perceive the stiffness of ECM including, among others, integrins-FA complexes linked to actin cytoskeleton, mechanosensitive channels and mechanosensitive transcriptional factors, such as the Hippo pathway effectors, which have been demonstrated to be able to modulate the myogenic transcriptional events in response to mechanical cues [14,179,182,197,198,199,200].

It worth highlighting that the satellite cell microenvironment, and similarly other stem cell niches, is a very dynamic compartment where the cell population that interacts with satellite cells, as well as the ECM stiffness, differ under homeostatic quiescent conditions with respect to repair/regeneration after a damage [1,174,178,201,202]. It is likely that in a physiological tissue repair after a focal damage, the modifications that occur in the microenvironment are compliant with stem cell proper functionality and favorable for assuring their successful myogenic differentiation, leading to the morpho-functional recovery of damaged muscle tissue as well as self-renewal to replenish the satellite cell pool. After tissue damage, immune cells (mainly neutrophils and macrophages) are recruited, transiently infiltrate the satellite cell microenvironment and, together with degenerated or necrotic myofibers, firstly convey biochemical signals for the activation of satellite cells. “Nursing” interstitial cells also contribute to modulate satellite cells activities [177,201]. With regards to the ECM immediately surrounding satellite cells, a focal muscle damage frequently causes its initial insult, followed by a further physiological degradation by MMPs [178], likely reducing the intrinsic ECM stiffness. This appears essential to allow both the migration of satellite cells and the homing of inflammatory cells to the site of injury and also the release of ECM-tethered pro-myogenic factors. On the other hand, it is worth saying that signals from such a damaged microenvironment mainly activate resident MSCs, fibroblasts and myofibroblasts. These cells are responsible for the collagen deposition forming a provisional contractile scar required to rapidly restore tissue integrity and preserve organ function [194,203]. Consequently, ECM stiffness undergoes a transient increase, in line with data in the literature [181,189,202,204], but however, it results still conductive for the accomplishment of satellite cell mediated-regeneration mechanisms. The scar will be ultimately removed and replaced by normal tissue. By contrast, in case of a persistent or extended damage or in the presence of a persevering inflammatory stimuli, it is possible to assist to a prolonged and excessive ECM accumulation and stiffening, compromising the achievement of tissue regeneration and leading to permanent tissue scarring [10,21,178,194,202,203].

### 5.2. Potential Impact of S1P-S1PR Signaling on Satellite Cell and Neighboring Interstitial Cell Functions and Interplay

Dynamic changes in S1P metabolism and signaling after a muscle damage or disease/myopathy have been documented, suggesting a role for this biolipid in influencing the endogenous mechanisms of tissue repair/regeneration [205,206,207,208,209]. It is conceivable that resident satellite cells may be target of circulating/systemic S1P. Indeed, the capability of S1P/S1PR mediated signaling pathway to modulate proliferation, migration and differentiation of different muscle progenitor/myogenic cells and muscle regeneration *in vivo* [206,209,210,211,212] and *ex-vivo/in vitro* [205,207,211,212,213,214,215,216,217,218,219,220,221,222,223,224] has been documented. In addition satellite cells, differentiated myofibers and interstitial cells, such as MSCs and macrophages, could also locally release S1P, which may act via S1PRs in a autocrine and paracrine manner for regulating cell behavior [205,225,226] (Figure 1).

In this line, a study of our research group demonstrated that the activity of SphK1, the kinase that catalyzes the formation of S1P, is required for the myogenic differentiation of C2C12 myoblasts. Indeed, SphK1 activity, SphK1 protein content and S1P formation were found to be enhanced in differentiating myoblasts. Moreover, it was also reported that myogenesis in SphK1-overexpressing myoblasts was abrogated by treatment with short interfering RNA specific for S1PR2 [218] suggesting that the locally synthetized S1P mediates the myogenic differentiation events after its release and interaction with receptor, thus indicating the pro-myogenic role of S1P/S1PR signaling.

To support the involvement of myofiber-released S1P and S1PR axis in myogenic events, recent findings demonstrated a down regulation of SphK1 activity, an increased expression of S1P transporter SPNS2 and S1PR2 in myotubes induced to atrophy by dexamethasone, thus indicating a negative role of S1P/S1PR2 axis in the maintaining of normal mature myotube phenotype [57].

Our research group has also demonstrated the capability of bone marrow-MSCs to secrete S1P. The released S1P represents an important factor by which MSCs exert their paracrine stimulatory effects on the proliferation of C2C12 myoblasts and satellite cells [227]. Interestingly, we have reported that autocrine release of S1P by MSCs and S1PR1 activation exert a trophic role in maintaining the ability of MSCs to modulate MMP-2 expression and function [228]. In the same study, we also demonstrated that MMP-2 expression and release are necessary not only for ECM degradation, but also for cytoskeleton reorganization and cell proliferation in both conditions of normoxia and hypoxia, which may resemble a damaged/regenerating tissue microenvironment. These results appear of particular interest when considering that all those cellular events may positively influence satellite cells behavior. Indeed, it is tempting to speculate that released MMP-2 may contribute to ECM degradation favoring both satellite cell migration, myoblast fusion, MSC recruitment (proliferation and migration) to the injury site and also the release of ECM-tethered pro-myogenic factor (such as HGF) [178,202,229] (Figure 1). Furthermore, the occurrence of a potential interaction cannot be excluded between MMP-2 and syndecan 4 [230], a cell-surface transmembrane heparan sulfate proteoglycan, whose regulatory role in satellite cell maintenance, activation, proliferation and differentiation has been documented [231,232].

On the other hand, a role of S1P/S1PR signaling in the modulation of the functionality of macrophages and other inflammatory cells potentially invading the damaged satellite niche, with potential repercussion on stem cell behavior, has been widely demonstrated [233,234,235,236].

### 5.3. How S1P-S1PR Signaling May Influence Satellite Cell Mechanotrasduction Events: Actin Cytoskeleton as a Key Target

Cytoskeleton, physically and/or functionally linked to different receptors or transcriptional factors acting as mechanosensors in satellite cells, seems to be the key player in mechanotransduction, being responsible for relaying and moderating transcriptional events in response to mechanical cues. It has been reported that in response to stiff environment cytoskeleton network undergoes dynamic changes mainly consisting in acto-myosin contractile stress fiber formation [10].

One of the referred mechanism by which stress fiber may regulate transcriptional events in satellite cells is related to the depletion of G-actin pool after stress fiber formation. Indeed, G-actin is able to sequester some transcriptional factors, namely myocardin-related trascritpion factor (MRTF), and is able to activate serum response factors, which are crucially involved in the expression of muscle specific genes [237]. When the monomeric G-actin becomes polymerized into F-actin, within stress fibers, MRTF are liberated and translocate to the nucleus [179,197]. In addition, contractile stress fibers, generating an increase of intracellular tension, may contribute to the opening of mechanosensitive channels, namely, Stretch Activated Channels (SACs), whose essential role in both early proliferative stage and subsequent differentiation of skeletal muscle myoblasts have been described [198,217,219,238,239].

Given that the impact of S1P/S1PR on actin cytoskeleton dynamics in different cells including muscle precursor cells has been widely demonstrated [228,240,241], it is reasonable to assert that this signaling is involved in modulation of the satellite cell mechanotransduction. In this context, our research group demonstrated that activation of S1P/S1PR signaling (Edg3/ S1PR3 and Edg 5/S1PR2) elicits robust cytoskeletal rearrangement in murine C2C12 myoblastic cells, mainly consisting in the formation of stress fibers and assembly of FAs through the activation of Rho-and phospholipase D (PLD)-mediated pathways. These events were associated with an increase of intracellular calcium transients propagating throughout the cytoplasm and nucleus as well as cell contractility [213,217,225,238,242,243,244]. Considering the reported role of calcium in mechanotransduction in the skeletal muscle tissue [245], the potential involvement of S1P/S1PR signaling in the mechanotransduction events in satellite cells via modulation of calcium transient cannot be excluded.

Interestingly, we also revealed that S1P-induced stress fiber formation is associated with the onset of plasmamembrane tension and the activation, in turn, of SACs [238,244]. The functional interaction between stress fibers and SACs is required for myogenic differentiation of myoblasts [217]. Indeed, both the disruption of actin cytoskeleton and the impairment of channel functionality hamper myogenesis.

Consistent with the assumption of the potential relevance of S1P/S1PR signaling in satellite cell cytoskeleton mediated-mechanotransduction events correlated to myogenic differentiation, we showed the ability of this signaling to promote the functional assembly at membrane of an essential component of SACs, namely transient receptor potential canonical channel (TRPC)1 by the induction of stress fiber formation. Concomitantly, S1P/S1PR signaling modulates also TRPC1 expression and the channel activity, which are required for skeletal myoblast differentiation [219].

Furthermore, we have demonstrated that a key regulatory protein of C2C12 myoblast differentiation elicited by S1P/S1PR signaling is connexin 43 (Cx43), one of connexin isoform forming gap junctions. Interestingly, beside its gap junction-dependent function, we demonstrated that Cx43 may regulate skeletal muscle differentiation *per se*, acting as adaptor protein. This gap junction- independent function requires the physical association of Cx43 with cytoskeletal protein (F-actin and cortactin) and the cyoskeleton remodeling, events that are both elicited by S1P [215]. Interestingly, Cx43/cytoskeleton interaction promoted by S1P during myoblast differentiation is dependent by S1P-mediated TRPC1 activation [246].

Finally, emerging studies indicate that tension of the actomyosin cytoskeleton is required for the regulation of YAP and TAZ, involved in the modulation of muscle stem cell function [14,200]. Therefore, it is enticing to speculate that the cytoskeleton remodeling elicited by S1P/S1PR may impact also on this nuclear relays of mechanical signals and therefore modulate satellite cell mechanotransduction (Figure 1). On the other hand, a cytoskeleton-indipendent regulation of the Hippo system/YAP/TAZ by the S1P/S1PR signaling in satellite cells as occurs in other cell types as stated in the Section 4.2, cannot be excluded and is worth investigating.

### 5.4. How S1P-S1PR Signaling May Influence ECM Stiffness

As far as we know, direct evidence of the contribution of S1P/S1PR signaling in the modulation of ECM stiffness in skeletal muscle with repercussion on satellite cell activities are lacking. Nevertheless, its postulation appears reasonable on the basis of data revealing the impact of this signaling on muscle ECM remodeling and on the capability of different cell types (likely occupying satellite cell microenvironment, such as the same satellite cells, fibroblasts, myofibroblasts, MSCs, and even inflammatory cells/macrophages) to synthetize and deposit fibrillar collagen and/or release MMPs. Along this line, a study from our research group, aimed to examine the effects of S1P on *ex-vivo* murine model of eccentric contraction (EC)-induced muscle damage, showed that the bioactive lipid attenuates the intermyofiber deposition of collagen in damaged muscles and up-regulates the expression of MMP-9, concomitantly stimulating the myogenic differentiation of the resident satellite cells [205], thus, confirming the influence of ECM stiffness on satellite cell behavior.

The role of S1P/S1PR signaling in modulating the functionality of fibroblasts from different tissues, including their ability to express and deposit collagen and synthetize MMPs, has been demonstrated [247] as well as the impact of the signaling on the modulation of the transition of fibroblasts towards myofibroblasts. Myofibroblast are the main cells responsible for formation of provisional scar and its removal during reparative process/tissue scarring and, in the worst conditions, of tissue fibrosis development [52,194,248,249,250].

For example, in the cardiac fibrosis context, SphK1 seems to be determinant [52,251]. In addition, it has been demonstrated that the expression of this enzyme is promoted by the well-known pro-fibrotic factor, namely Transforming Growth Factor (TGF)-β, and that the enzyme mediates the upregulation of TIMP-1. Moreover, the SphK1 silencing results in the inhibition of collagen production elicited by TGF-β [252,253]. Notably, the neutralization of extracellular S1P by a specific antibody, robustly decreases TGF-β-stimulated collagen synthesis, indicating that the “inside-out” S1P signaling, upon SphK1 activation, may play a role in the pro-fibrotic response [253]. However, at this point, it is worth mentioning that TGF- β may act not only as a pro-fibrotic agent, it might also positively influence stem cell recruitment and fate determination and ultimately regeneration and homeostasis of different adult tissue [254]. In line with this, an increase of the expression of this factor and of its receptor (type II receptor) in pancreatic acinar, ductal epithelial and islets cells during the early phases of spontaneous pancreatic recovery after ischemia/reperfusion (I/R)-induced pancreatitis in rats, has been observed [255,256]. The involvement of S1P/S1PR signaling in modulating such “non-canonical” role of TGF-β in muscle tissue regeneration [257] is worth to be investigated.

It is tempting also to speculate that the contribution of S1P/S1PR signaling on ECM stiffness may be linked to its involvement in the regulation of the expression and release of collagen and MMPs in MSCs, based on the following recent observations: 1) S1P/S1PRs (including S1PR1/R3 but not S1PR2) have been proven to play an important role in the regulation of collagen type I/III expression in mouse and human bone marrow-derived MSCs and on the acquisition of myofibroblastic phenotype by these cells [258,259] and 2) S1P/S1PR1 is required for the gelatinolytic activity of MMPs and for MMP-2 expression/function in murine bone marrow-derived MSCs [228] (Figure 1). Finally, an impact of S1P/S1PR signaling on matrix degradation and remodeling capacity of macrophages can be supposed [25,30,235,260,261,262,263].

## 6. S1P/S1PR and Modern Strategies for Regenerative Medicine

The modern strategies for regenerative medicine are based on the use of instructive biomaterials, such as hydrogels or polymer scaffolds, which help in the specific delivery of cells or in forcing cells to adopt a specific shape. Because of the peculiar functions of S1P in modulating the cell response to mechanical stimuli described in this review, the possibility to localize or modulate the S1P content may be a useful experimental approach to recruit stem/progenitor and effector immune cells in specific location [264], while influencing their behavior.

The advantages of the utilization of S1P in regenerative medicine are extensively revised by Marycz et al. [265,266]. Zheng et al. [267] were the first to show that material-controlled release of S1P could be clinically relevant. In fact, Jurkat leukemic T cells overexpressing the receptor subtypes Edg-2 and Edg-4 were able to differently migrate through a layer of Matrigel on a 5-um pore polycarbonate filter upon anti-Edg-4 or anti-Edg-2 R antibodies, treatment providing the evidence of a receptor-selective mechanisms for regulation of T cell recruitment and immune contributions.

Many studies demonstrate the utility of S1P to functionalize biomaterials. In particular S1P can be loaded on poly(lactic-co-glycolic acid) (PLGA), encapsulated within PLGA thin films or temporally delivered from porous hollow cellulose fibers. S1P loaded onto functionalized biomaterials, was found to enhance the osteogenic differentiation of human adipose-derived multipotent stromal stem cells [266] or to potentiate the osteogenic capability of rat bone marrow stromal cells *in vivo* when the bioactive lipid is coated on β-Tricalcium phosphate scaffold [268]. Other biomaterial systems, such as alginate hydrogels, have been used for controlling S1P release in order to create a local continued concentration gradients able to favor *de novo* vascularization [269]. Moreover, it has been documented that S1P-loaded on microparticles of PLGA improves blood flow recovery [270] and promotes short-term enlargement of arteriolar and venular diameters [271].

The biomimetic release of S1P in cultured MSCs can also be performed with chitosan-gelatin scaffold functionalized with microspheres that offers higher stability [272]. To enhance the stability of S1P effect, the bioactive lipid has been also co-delivered from PLGA films together with a S1P lyase inhibitor, which limits the degradation of the bioactive lipid, thus substantially increasing its local tissue concentrations over time [264].

Similarly, agonists/antagonists of S1PR have been loaded on specific matrix. For instance, hydrogels containing micelles of the S1PR1 agonist, SEW2871, were demonstrated to enhance macrophage migration in vitro as well as *in vivo* [273]. FTY720-loaded nanofibers have been successfully utilized to increase bone regeneration ability, reduce vascularization and local inflammation in a rat model, providing evidence for the use of S1P agonist in craniofacial defect recovery [274]. In addition, FTY720 delivered in PLGA thin films resulted in a local recruitment of factors and monocytes in inflamed and ischemic tissues [275].

Finally, the delivery of humanized anti-S1P monoclonal antibodies was found to be useful for limiting the microvessel tube formation by human brain ECs obtained from patients with macular degeneration and plated on Matrigel [276].

It is important to emphasize that the integration between biochemical factors together with biomechanical stresses become crucial points to be considered when studying the ability of stem cells to differentiate in specific cell lineage. In order to overcome the limits correlated to the traditional methods of culturing cells mainly on stiff substrates, many novel technologies taking into account these considerations are emerging in tissue regeneration and regenerative medicine field. The use of three-dimensional (3D) culture conditions, especially cell-seeded scaffold systems, allows the cells to reach a cell shape closer to the one of their natural environment [277]. Notably, the 3D printing technologies, used with biocompatible materials and stem cells, may create a field of 3D bioprinting for artificial organ printing [278], possibly opening new interesting windows in regenerative medicine approach.

## Figures and Tables

**Figure 1 ijms-20-05545-f001:**
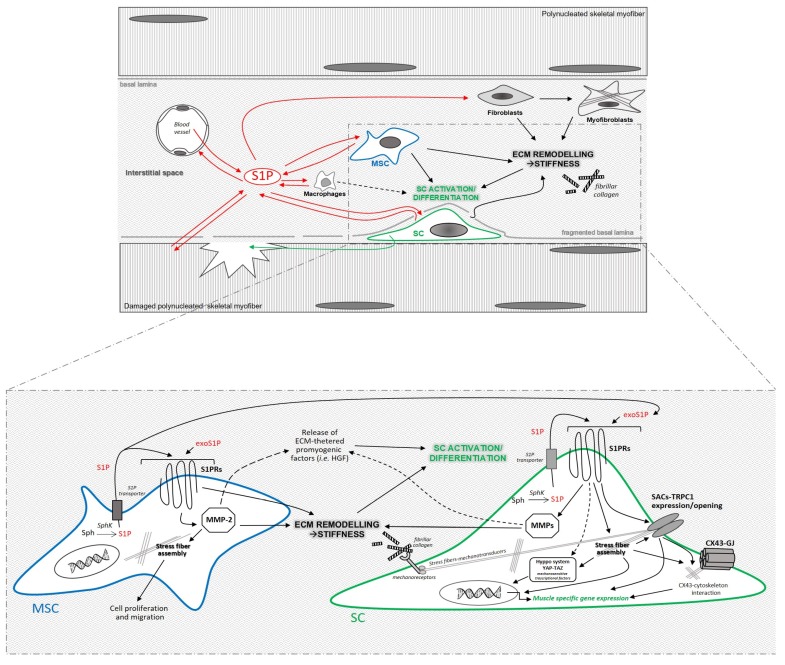
Drawing representing the potential influence of S1P released in the context of damaged skeletal muscle tissue by blood and different cells, via S1PRs in the regulation of mechanobiology of satellite cell (SC) niche, namely, SC mechanotransduction events in response to ECM stiffness and the modulation of ECM stiffness. The interplay between MSCs and SCs in such context has been highlighted. Abbreviations not present in the text: exogenous S1P (i.e., not synthetized and released by the target cell), exoS1P; gap junctions, GJ. Dotted lines: unknown pathways.

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
