# Peer review of "Sphingosine 1-Phosphate (S1P)/ S1P Receptor Signaling and Mechanotransduction: Implications for Intrinsic Tissue Repair/Regeneration"

_ijms, 2019, doi:10.3390/ijms20225545_

Round 1

Reviewer 1 Report

    It is interesting and intriguing review article, that you can't stop reading before its end. After reading this article submitted to me for review, however, a small comment came to my mind. The described suggested changes in the text lead to a better understanding of the theme and will increase readers' interest in this topic.

    The authors write: ”In HeLa cells, it has been documented that S1P, specifically formed by SphK2 and not by SphK1 activity, is essential for cell invasion elicited by epidermal growth factor [147]” and in another place: “It has been indeed demonstrated that SphK1 is induced by Transforming Growth Factor (TGF)-β and mediates TIMP-1 upregulation, and that siRNA against SphK1 inhibited TGF-β-stimulated collagen production [251,252]”. The authors at this point should mention the role of TGF in other organs in the context of the physiological relationship of inflammation. For example the presence of FGF, VEGF, PDGF-A and TGFbeta RII is modified in the course of I/R-induced acute pancreatitis. Maximal content of FGF, VEGF and TGFbeta RII has been observed in early stage of pancreatic regeneration (PMID: 15613744). Overexpression of transforming growth factors (TGF) in acute pancreatitis suggested that these substances play an important role in pancreatic repair and remodeling but the contribution of epidermal growth factor (EGF), that is well known to promote cell growth and regeneration. The expression of TGF-beta1 is biphasic with an initial increase probably related to pancreatic damage and inhibition of cell proliferation and with the later phase of increase accompanied by the stimulation of the synthesis of extracellular matrix components (PMID: 9586822).

    Authors write „Also the systemic milieu has to be included, comprising molecular and cellular signals deriving from the entire muscle belly along with neighboring skeletal muscles and bones, immune cells and circulating factors [173]”. A wide range of external and internal factors are required for proper muscle metabolism and contraction, especially gastrointestinal hormones. It should be mentioned (PMID: 25716961).

    Authors must improve the editing of the literature (spaces between individual items cited).

Author Response

Answer to Reviewer 1.

We thank the reviewer for his/her positive evaluation of our manuscript. Accordingly, we revised the text including new sentences and new references (page 10, lines 492-493; page 15, lines 680-694) and revising the editing of the literature.

Reviewer 2 Report

The authors present an extensive review of the literature regarding the role of S1P/S1PR signalling with regards to tissue regeneration/repair. This is a very well written review that systematically introduces the background on S1P/S1PR signalling and their various roles. The authors also have done a decent job with the graphical representation.

Author Response

Answer to Reviewer 2.

We thank the reviewer for his/her positive evaluation of our manuscript.